# Consider the Source: The Impact of Social Mixing on Drylot Housed Steer Behavior and Productivity

**DOI:** 10.3390/ani13182981

**Published:** 2023-09-21

**Authors:** Courtney L. Daigle, Jason E. Sawyer, Reinaldo F. Cooke, Jenny S. Jennings

**Affiliations:** 1Department of Animal Science, Texas A&M University, College Station, TX 77843, USA; reinaldo.cooke@tamu.edu; 2King Ranch^®^ Institute for Ranch Management, Texas A&M University-Kingsville, Kingsville, TX 78363, USA; jsawyer@eastfoundation.net; 3Texas A&M AgriLife Research, Texas A&M University, Bushland, TX 79012, USA

**Keywords:** social mixing, feedlot, steer, cattle, comingling, welfare

## Abstract

**Simple Summary:**

Social mixing, a critical component of contemporary beef cattle management, can cause psychosocial stress and typically occurs simultaneously with other stressors (e.g., weaning, transportation, etc.), leaving the independent impact of social mixing on cattle welfare unknown. To begin to disentangle this web, two different sources of genetically similar steers were either socially mixed or housed with individuals from their source herds, and their productivity and behavior were monitored. Social mixing did not affect average productivity in pens, yet the impacts that were observed at the individual steer level indicated that social mixing negatively affected the productivity of steers from one source but not from the other. Irrespective of social mixing, sources differed in the amount of time per day they spent ruminating and drinking. Social mixing was not universally detrimental to cattle welfare, and group-level evaluations can mask welfare challenges experienced by individuals. Furthermore, the source of cattle may have the greatest impact on steer performance, regardless of the social mixing treatment employed.

**Abstract:**

Cattle are a social species in which social mixing can induce physical and psychosocial stress; however, the impact of social mixing on cattle welfare is unknown. Two different sources of genetically similar Angus crossbred steers were transported to the same feedlot and assigned to a pen where they were either socially mixed or housed with individuals from their source herds. Social mixing did not impact average daily gains in pens, feed intake, or feed efficiency; pens of socially mixed steers were more active. Sources differed in their responses to social mixing. One source was unaffected, whereas social mixing negatively impacted productivity for the other source. Irrespective of social mixing, the sources differed in the amount of time per day they spent ruminating and drinking. Group analyses indicated that socially mixing two sources of feedlot steers did not negatively impact group productivity, yet the impacts that were observed at the individual level suggest that prior experiences may influence their ability to cope with social stress, emphasizing the importance of early-life experiences to long-term welfare and productivity. Social mixing was not universally detrimental to cattle welfare, and the source of cattle may have the greatest affect on their performance regardless of whether a social mixing event has occurred.

## 1. Introduction

Cattle are gregarious creatures that live in matriarchal groups interconnected by long-term non-familial social bonds [1]. The modern beef production system forces many calves to develop their social skills at a young age. Cattle in the auction system are typically young or recently weaned and may experience multiple social mixing events within a short period of time. Calves that traverse the auction system have been observed to have an increased risk of developing pneumonia at the feedlot [2], lighter body weights upon arrival, reduced ADG, increased morbidity rates, and shorter latencies to first respiratory treatment compared to calves transported directly to a feedlot from a ranch [3]. Calves arriving in a single truckload may have come from different sources and have had different journeys. In Western Canadian auction markets, calves from as many as 20 to 30 producers may be on the same truck [2]. These journeys vary not only with respect to the herd of origin but also in terms of rigor and valence. Each calf will have a unique suite of experiences as they interact with humans and are socially mixed during their transition from the ranch to the feedlot.

As cattle enter the finishing phase of the beef production system, they may be sorted and socially mixed into homogenous groups based on certain attributes (e.g., size, sex, and breed) [4,5] to facilitate efficient and mindful husbandry (e.g., in terms of feed delivery, distance to loading facilities, etc.). Social mixing is common, routinely occurs in close temporal proximity to other stressors (e.g., transportation and weaning), and is often implicated as a source of negative outcomes. However, the complications surrounding the study of social mixing among cattle leave the independent impact of social mixing on cattle behavior, resource use, and productivity poorly characterized. 

Social mixing is a psychosocial stressor that challenges cattle welfare by catalyzing social instability [6], resulting in destabilized social structures [5] that alter the social and competitive environment [7] within a group. This process is characterized by increased levels of aggression and resource defense [8] and reduced rates of rumination [9] as cattle re-establish their social hierarchy. Consequently, some cattle may experience poor welfare associated with a lower social status as they may have a limited ability to access and use resources (e.g., feeder, water, etc.), which can negatively impact growth and productivity. However, the impact of social mixing has been observed to be transitory [10], and the degree of the impact on dairy cattle productivity has been observed to vary at the individual level [9]. 

There is limited knowledge regarding the impact of social mixing on beef cattle welfare and the intricacies of beef cattle social dynamics [11]. To address this gap, the impact of social mixing, independent of other stressors (e.g., transportation, weaning, diet changes, etc.), on productivity, efficiency, social behavior (e.g., affiliative and agonistic interactions), resource use, herd synchrony, and rumination behavior was evaluated using two sources of steers of similar breed compositions and management histories. 

## 2. Materials and Methods

This study took place over a 42 d period in January and February 2019 at the Texas A&M AgriLife Research Feedlot in Bushland, Texas. Throughout the duration of the study, climatic conditions varied: maximum daily temperatures ranged from −6 °C to 15 °C, and relative humidity (RH) ranged from 11% to 96%. Only ~19% (8 d) of the days within the study had average daily temperatures below 0 °C, while ~81% (34 d) had minimum daily temperatures below 0 °C. Precipitation occurred on ~12% (5 d) of the days, with only one day witnessing snowfall. Total precipitation not including snowfall ranged from 0.03 cm to 0.45 cm, with an average of 0.15 cm/day of rainfall, while only 0.76 cm of snow accumulated. Average daily temperature humidity index (THI) ranged from a daily low of 31.4 ± 1.08 to a daily high of 54.60 ± 1.32.

### 2.1. Animals and Housing

In total, 96 steers (¾ Angus × ¼ Nellore) of similar genetic composition from two geographically separate Texas A&M AgriLife operations (*n* = 48 from SOURCE_A (College Station, TX, USA) and n = 48 from SOURCE_B (McGregor, TX, USA)) were used in this experiment. Before weaning, the cow–calf pairs from SOURCE_A grazed in pastures that were near highways and were handled frequently throughout the rearing period as part of a research project involving the cows. The cow–calf pairs from SOURCE_B ranged extensively and were handled only during calving, pre-weaning, and weaning. Thus, the handling experiences and consequential social disruptions of the calves before weaning differed in intensity (e.g., frequency of handling events within a period of time, time separated from dam during each handling event, etc.), duration (e.g., time to muster, pre-processing holding time, and distance to handling facility), and human–animal dynamic (e.g., number and diversity of handlers, presence or absence of a procedure during a handling event, etc.) between these two operations when a human–animal interaction was required. Calves from both sources were weaned simultaneously on d −82 and backgrounded at their home operation following the same protocols (Figure 1).

On d −39, each source was transported (Source_A: 842 km; Source_B: 735 km) via separate trucks to the Texas A&M AgriLife Research Feedlot in Bushland, Texas. Steers were housed separately according to their sources without visual or tactile contact in order to separate transportation and relocation stressors from social mixing. Five days after arrival (d −34), cattle were weighed and affixed with an ear tag containing a three-axis accelerometer (Allflex Livestock Intelligence, Madison, WI, USA) to continuously record daily activity and total time per day spent ruminating. At the same time, cattle were administered a *Mannheimia haemolytica* leukotoxoid vaccine (Presponse SQ, Boehringer Ingelheim Vetmedica, St. Joseph, MO, USA), an anthelmintic (Dectomax, Zoetis), and a growth-promoting implant (Revalor-IS, Merck Animal Health, Kenilworth, NJ, USA). Steers were fed a standard growing ration (Table 1) once daily from the day of arrival (d −39) through to the last day of the study (d 42) and had ad libitum access to water. Feed bunks were managed so that less than 0.45 kg of feed remained in the bunk the following day.

### 2.2. Data Collection

Before being sorted into assigned pens on d 0, steers were weighed and fitted with colored collars for individual identification in video recordings. Exit velocities (EVs) (as described in [12,13]) were measured across a 1.8 m distance using electronic timers (FarmTek, Inc., Wylie, TX, USA). Body weights (BW) were collected on d 0, 24, and 42, and EVs were collected on d 0 and 42. 

Cattle behavior in the home pens was recorded from 0800 to 1800 on d 0, 1, 2, 3, 4, 5, 6, 7, 8, 9, 14, 15, 16, 21, 22, 23, 28, 29, 30, 35, 36, and 37 relative to social mixing using a closed-circuit video camera recording system (Points North Surveillance Systems; Auburn, ME, USA). Using 10 min instantaneous scan sampling (61 scans/pen/day; as reported in [14]), the total number of steers lying, eating, and performing other behaviors within a pen were recorded (Table 2). Continuous observations recorded the frequency that each steer initiated allogrooming and agonistic behaviors (i.e., feed bunk displacements, waterer displacements, headbutts, and mounts) in addition to the frequency and duration of drinking bouts (Table 2). In the present study, cattle were only recorded as drinking if their snouts were in the water, water tension was visibly disrupted, and steers were not actively playing with any feature of the waterer. Thus, only the duration and frequency with which cattle were actively drinking were recorded.

All behavior data were collected by 30 trained observers using BORIS (version 7.9.15; [15]. Inter-observer reliability between both observer and trainer, and among observers, was assessed before data collection in which all participants scored the same pen in a video and Pearson’s Rho was calculated. All inter-observer reliability tests produced an R-value greater than or equal to 0.95, indicating excellent agreement between observers. Inter-rater reliability was re-evaluated for each observer each time they completed data collection on 20 h of video (i.e., two days of video).

### 2.3. Statistical Analyses

One steer (n = 1, SOURCE_A) was removed from the study on d 16 due to the presence of an unknown strain of a viral wart. A replacement steer was not added to the pen to avoid confounding the experiment through having an additional day where social mixing occurred. Behavior data were not collected for two pens (n = 1 SOURCE_B and n = 1 MIX) after d 3 due to technical issues that resulted in faulty video transmission.

Data were analyzed using each pen as an experimental unit and the Satterthwaite approximation to determine the denominator degrees of freedom for tests of fixed effects. Shapiro–Wilk and Levene tests were used to test the data for normality and homogeneity of variance, respectively. 

Scan samples were arcsine-square-root-transformed, and drinking behaviors, social behaviors, ADG, and G:F were all log-transformed to achieve normality (W > 0.90) and homogeneity (*p* ≥ 0.10) of variance. Results were back-transformed for description. 

The Shannon–Weiner Diversity index [16] was used to calculate herd synchrony in this study, wherein the calculated index values can be influenced by the number of behaviors measured [17]. As the number of behaviors measured increases, the potential for individuals to be recorded as performing different behaviors also increases. As the number of individuals performing different behaviors rises, the Shannon–Weiner Diversity index increases, indicating lower herd synchrony and reduced group cohesion.

Three-day moving averages were calculated for both rumination and activity times, as stressors are expected to manifest in changes in rumination and activity up to three days following the occurrence of a stressor [18].

Proportions of cattle performing core behaviors (e.g., lying, eating, etc.) as well as Shannon’s D were averaged according to week, and the impacts of treatment, week, and their interaction were analyzed using a Generalized Linear Mixed Model (PROC GLIMMIX) in SAS (SAS Inst. Inc., Cary, NC, USA). The random effects included the pen-within-treatment-by-day interaction, the observation-within-day-by-week interaction, and day within week. 

The impact of treatment on productivity was analyzed using a Generalized Linear Mixed Model (PROC GLIMMIX) in SAS (SAS Inst. Inc., Cary, NC, USA). Pen within treatment was the random effect. 

The impact of treatment, time, and their interaction on the daily totals of social behavior was analyzed using a Generalized Linear Mixed Model (PROC GLIMMIX) in SAS (SAS Inst. Inc., Cary, NC, USA). Pen with the treatment by day interaction was the random effect. 

Evaluating the impact of a treatment using only the group average can mask existing welfare concerns because individuals within the same group may have opposite responses to the same scenario. Thus, an individual animal’s response, in addition to group-level assessments, needs to be considered when evaluating the impact of these types of management practices. The authors acknowledge that the results of an individual-level analysis of animals that are housed in groups is subject to pseudo-replication due to violations of the independence among observations that are required to meet the assumptions of the statistical model. While this approach incurs a risk of Type I and Type II errors, this evaluation is suitable for truly understanding the impact of social mixing on cattle welfare. Thus, both statistical approaches were taken in this study, but the individual-level results should be interpreted with caution.

To examine the impact of social mixing on steers from different sources on data that were recorded on an individual-steer basis, each steer was assigned to one of four source–treatment combinations: (1) SOURCE_B-NOMIX, (2) SOURCE_B-MIX, (3) SOURCE_A-NOMIX, and (4) SOURCE_A-MIX. Behavior (rumination, activity, and drinking) and production responses (e.g., ADG, DMI, and G:F) were analyzed using a Generalized Linear Mixed Model (PROC GLIMMIX) in SAS (SAS Inst. Inc., Cary, NC, USA) using treatment as the fixed effect and steer within pen × treatment as the random variable. 

For all analyses, when overall differences were detected, post hoc analyses were conducted with pair-wise t-test comparisons of least-squares means using a Bonferroni adjustment. Values reported are least-squares means, and significance was declared at *p* ≤ 0.05.

## 3. Results

### 3.1. Productivity

Social mixing did not impact (*p* ≥ 0.05) ADG, DMI, or G:F (Table 3), i.e., no differences were observed between the three groups, although pens containing mixed-origin steers had the numerically lowest ADG. No incidences of morbidity or mortality were observed during the experiment.

When productivity metrics were analyzed to determine the impact of the source–social mixing treatment combination (Table 4), differences between the treatment–source combinations were more readily observed. Differences between the treatment–source combinations tended to be observed for ADG (*p* = 0.10) and G:F (*p* = 0.07). The SOURCE_B-NOMIX, SOURCE_A-MIX, and SOURCE_A-NOMIX steers had ADG values that were 12%, 12.6%, and 10.7% greater, respectively, than the SOURCE_B-MIX steers. Further, the SOURCE_B-NOMIX, SOURCE_A-MIX and SOURCE_A-NOMIX steers were numerically more efficient than the SOURCE_B-MIX steers.

### 3.2. Core Behaviors and Herd Synchrony

The proportion of steers eating at any given time (Figure 2a) was impacted by both the treatment (*p* = 0.005) and the week (*p* = 0.029). A larger proportion of the SOURCE_A steers were observed eating compared to the SOURCE_B and MIX steers; these two treatments did not differ (Table 5). Neither treatment nor week influenced the proportion of steers lying simultaneously (Figure 2b). Treatment tended to influence the proportion of steers performing other behaviors (*p* = 0.059), with a larger proportion of steers performing other behaviors in the MIX group compared to the SOURCE_B and SOURCE_A. groups. However, these proportions did not change over time (Figure 2c). These differences are reflected in the observed degree of herd synchrony.

Herd synchrony was influenced by treatment (*p* < 0.0001), wherein the steers in the SOURCE_B pens were more synchronous (e.g., less diverse) in their behavior than those from either the MIX or SOURCE_A pens (Figure 2d). These metrics could not be analyzed for the source–treatment combination, as the data were collected on groups of animals and since data on individual animals were unavailable.

### 3.3. Rumination and Activity

The daily length of time (min/d) the cattle spent either being active (Figure 3a) or ruminating (Figure 3b) varied across time (*p* < 0.0001) and differed by social mixing treatment (Table 5). The steers from SOURCE_A spent 18.7 more minutes per day ruminating compared to the MIX steers (*p* = 0.001) and 29 more minutes per day than the SOURCE_B steers (*p* < 0.0001). The steers in the MIX pens spent more time active than the steers in the SOURCE_A (*p* < 0.0001) and SOURCE_B (*p* < 0.0001) pens. The steers in the SOURCE_A pens were more active than those in the SOURCE_B pens (*p* < 0.0001). Irrespective of social mixing treatment, the steers ruminated least on d18 and most on d30 (*p* < 0.0001). Steers were the most active on d 4 and least active on d37 (*p* < 0.0001).

When the impacts of the different social mixing treatment–source combinations (SOURCE_B-NOMIX, SOURCE_B-MIX, SOURCE_A-NOMIX, and SOURCE_A-MIX) on the amount of time per day spent active were evaluated (Table 4), no differences between the treatment–source combinations were observed, yet activity levels changed over time (Figure 3c; *p* < 0.0001). An interaction between the social mixing treatment–source combination and day was observed for rumination (Figure 3d). The SOURCE_B-MIX steers spent the least amount of time ruminating on d 22 (*p* = 0.04), 23 (*p* = 0.04), 24 (*p* = 0.01), 25 (*p* = 0.02), and 26 (*p* = 0.006). Regardless of the type of treatment–source combination, steers ruminated least on d18 and most on d30 (*p* < 0.0001).

### 3.4. Drinking Behavior

#### 3.4.1. Drinking Bout Frequency

The frequency of drinking bouts was impacted by the social-mixing-treatment-by-day interaction (*p* < 0.0001). Treatments differed in relation to the steers’ drinking frequency on days 1, 2, 3, and 8. Steers from SOURCE_B drank less than steers from SOURCE_A and MIX on d 1 (*p* = 0.0007) and 2 (*p* = 0.005). This trend was reversed on d3, on which steers from SOURCE_B drank more frequently than those from SOURCE_B and MIX (*p* = 0.008). Steers in the MIX pens drank less frequently than those from SOURCE_A or SOURCE_B on d8 (*p* = 0.007).

When drinking frequency was analyzed to determine the impact of the source–social mixing treatment combination (Table 4), an interaction between treatment combination and time was observed (*p* < 0.0001). Steers from SOURCE_B-NOMIX drank less frequently than steers from SOURCE_A-NOMIX and SOURCE_A-MIX on d1 (*p* = 0.028) and d2 (*p* = 0.008). Steers from SOURCE_B-MIX drank less frequently than steers from SOURCE_A-MIX on d8 (*p* = 0.024). The source–social mixing treatment combination impacted (*p* = 0.04) drinking frequency: steers from SOURCE_B-MIX drank less frequently than those from SOURCE_A-MIX.

#### 3.4.2. Daily Drinking Duration

The amount of time steers spent drinking was impacted by the treatment by day interaction (Figure 4b; *p* < 0.0001). The steers from SOURCE_B spent less time drinking than the steers from MIX or SOURCE_A on d1 (*p* = 0.001). However, MIX steers spent more time drinking on d2 (*p* = 0.005) and the least amount of time drinking on d3 (*p* = 0.024). On d8, SOURCE_A steers spent more time drinking than either MIX or SOURCE_B steers (*p* = 0.002).

When the amount of time steers spent drinking per day was analyzed to determine the impact of the source–social mixing treatment combination, an interaction between treatment and time was observed (*p* < 0.0001). However, after a Bonferroni correction, no differences on any day were observed between the treatment–source combinations.

#### 3.4.3. Drinking Bout Duration

The length of time the steers spent drinking per drinking bout was impacted by the treatment by day interaction (Figure 4b; *p* < 0.0001). Differences between treatments were observed for d 2 (*p* = 0.03), 3 (*p* = 0.0002), 4 (*p* = 0.03), and 8 (*p* = 0.04). Steers in the SOURCE_A pens had the shortest drinking bout duration on d 2 and the longest drinking bout duration on d 3. Steers in the MIX pens had the shortest drinking bout durations on d 4, and SOURCE_B steers had the longest drinking bout durations on d 8.

When drinking bout durations were analyzed with respect to the impact of the source–social mixing treatment combination, an interaction between treatment combination and time was observed (*p* < 0.0001). Differences between the source–treatment groups were observed for d 2 (*p* = 0.01), 3 (*p* < 0.0001), 4 (*p* = 0.006), and 8 (*p* = 0.033).

#### 3.4.4. Waterer Displacements

The daily frequency of waterer displacements was not impacted by treatment, day, nor their interaction. These metrics could not be analyzed for the source–treatment combination, as the data were collected on groups of animals and data on individual animals were unavailable.

### 3.5. Social Behavior

Treatment (*p* = 0.02) and time (*p* < 0.0001), but not their interaction, impacted the frequency of allogrooming bouts (Figure 5a). More allogrooming was observed in the MIX and SOURCE_A pens compared to the SOURCE_B pens. Steers engaged in the least amount of allogrooming (count/steer) on d 1 compared to all other days (*p* < 0.01) except d 8 (*p* = 0.07).

Daily frequencies of feedbunk displacement (Figure 5b), waterer displacement, head butting (Figure 5c), and mounting were not impacted by social mixing treatment. Steers from SOURCE_B tended (*p* = 0.09) to perform fewer long headbutts compared to the steers in MIX and SOURCE_A pens.

The frequencies of prolonged headbutts (Figure 5d; *p* = 0.0002), head butts (*p* < 0.0001), and feedbunk displacements (*p* = 0.02) changed over time. The most headbutts, long headbutts, and feedbunk displacements were observed on d8 compared to d0. Mounting frequency was not impacted by time or treatment.

## 4. Discussion

Socially mixing two sources of cattle may not be universally stressful for all cattle involved. The results of this research emphasize the importance of early-life experience for long-term cattle welfare, as calves from different sources differed in their performance at the feed yard. Further, the results from this study highlight the need to assess welfare at the individual level, as the impact of a social mixing treatment on cattle welfare differed depending on the metric evaluated and the source herd. The differences between sources in their patterns of core (i.e., ruminating and drinking) and social (i.e., allogrooming and prolonged headbutts) behaviors demonstrate that some source effects persist even after a social mixing event because these behaviors are inflexible in nature and less susceptible to social facilitation. Some, but not all, of the behaviors were influenced by source, suggesting that some behavioral patterns are established early in life and are relatively inflexible (e.g., drinking bout frequency), uninfluenced by social mixing, and require extreme stressors to alter (e.g., heat stress). This contrasts with behaviors that are sensitive (e.g., eating, rumination, etc.) to current conditions (e.g., diet, health status, etc.) and will change accordingly. This means that regardless of social mixing, some behaviors may change while others may not. The severity of these changes may be influenced by the experiences that calves have had before the mixing event, thus highlighting the importance of pre-weaned calf management for long-term cattle welfare and producer profitability. Understanding the history and experiences of calves that are to be mixed can facilitate decision making as to whether to implement this management practice.

The differences observed between the group-level and individual-level analyses highlight the complex interplay between individual welfare and group assessments. The experience of animal welfare takes place at the level of the individual [20]; however, animal evaluation and welfare assessment typically occur at the group level. Thus, the impact of social mixing on cattle welfare may be masked by the evaluation of group averages [21], potentiating a situation in which individuals experiencing welfare challenges may be overlooked. Further, the source-relevant experiences (e.g., weaning management, transportation distance, etc.) may provide insight into whether a group of cattle may or may not be impacted by a social mixing event. Many of the differences observed in this study were primarily conditioned on source, emphasizing the importance of understanding the inherent differences among cattle sources and how these differences may impact their capacity to cope with multiple types of stressors, including social mixing, later in life.

### 4.1. Impact of Social Mixing on Productivity

Group-level ADG, DMI, and G:F during the 42-day experiment were not impacted by social mixing. These results contrast with those reported by Arthington et al. [19] and Step et al. [10], who each found that groups of socially mixed and non-socially mixed calves differ in terms of DMI and ADG, respectively. However, the calves in both studies had different previous experiences before mixing, and the researchers had limited knowledge regarding their weaning, health, transportation, and management history, all of which are factors that could have confounded the interpretation of a calf’s response to commingling [3,22]. The influence of these confounding factors is emphasized by the results of this study, as many of the factors that have confounded previous research evaluating social mixing among beef cattle were minimized. 

The impact of social mixing on ADG differed between the two sources. Steers from SOURCE_B tended to be negatively impacted by comingling, whereas the steers from SOURCE_A were not. SOURCE_B steers had greater EV, suggesting that SOURCE_B steers may have had inherently stronger stress responses (e.g., proactive temperament) to any stressor, including social mixing, resulting in a subsequent negative impact on productivity and efficiency. The early-life experiences of cattle can have lifelong implications for their ability to cope with stress, including social stress [23], and this phenomenon can be observed across the animal kingdom. Among rats and mice, differences in an individual’s coping style are associated with susceptibility to autoimmune disease and stress-related health disorders [24,25]. Regarding pigs, differences in temperament and coping style were associated with immune functioning [26,27], carcass quality [28], maternal behavior, and reproductive success [29,30]. Among cattle, coping style varies at the individual level, and exit velocity (EV) is generally considered to be indicative of coping style [31]. Faster EVs have been negatively correlated with growth and meat quality [32,33,34], immune function [35,36,37], and pregnancy rates [38] and positively correlated with cortisol levels during handling [39]. Therefore, the steers from SOURCE_B might have had a reduced physiological capacity to cope with social stress, yet the social mixing treatment was not stressful for cattle from all sources.

Ontogeny, age, and genotype are known to impact the development of an individual’s temperament and their ability to cope with different types of stressors later in life [40]. Steps were taken to mitigate variation between the herds regarding early life experiences after weaning; however, the social dynamics within each source’s maternal herd might have resulted in calves with different social knowledge sets and social experiences before the commencement of this study. Further, calves from both sources experienced different levels of habituation to humans and human-managed environments. Before weaning, steers from SOURCE_A were handled more frequently and by a wider variety of people, and they spent more time in the holding pens during each handling event (longer duration), so they were likely more habituated to the more frequent and spatially close human–animal interactions that occur in the feedlot environment. This contrasts with the SOURCE_B calves that ranged extensively and were minimally handled before weaning. While the cattle were related and of a similar genetic type, they might not have been identical. Further variation in the management of each source’s maternal herd before weaning might have resulted in calves that vary in their exposure and habituation to humans and handling later in life. 

### 4.2. Impact of Social Mixing on Behavior

Rumination is a cattle comfort behavior [41,42,43] that can be influenced by feed intake and diet. In the present study, SOURCE_B steers ruminated less than steers from either the SOURCE_A or MIX pens. Various bacteria genera have been linked to feed efficiency traits that can influence rumen function [44,45,46], and rumen microbial composition is known to vary significantly depending on diet and geographic location [47]. The sources in this study originated from two different geographic locations, potentially resulting in exposure to different maternal microbiota, vegetation during rearing, and diets before weaning, all of which are known to alter rumen microbiomes [48]. Thus, differences in rumen microbiomes might have led to differing rumination patterns between the two sources, and they may serve as a better reflection of the inherent differences between sources compared to a behavioral response to social stress. 

Very little is known regarding the drinking behavior of cattle, yet recent advancements in technology designed to monitor drinking behavior (e.g., real time location systems [49], motion detectors, accelerometers, and weighted triggered plates [50]) have demonstrated that drinking behavior is variable between individuals [51] and changes in response to current environmental conditions [52]. However, while these technologies are designed to quantify the amount of time that cattle spend in proximity to a water source, they cannot detect actual water consumption. Cattle may aggregate around a water source in groups, even if they are not actively drinking [53], which can confound data collected regarding drinking behavior. 

It was observed that source influenced drinking behavior when the source–treatment combination was examined. Steers from SOURCE_A spent more time drinking per day and drank more frequently than SOURCE_B. The drinking bout durations averaged ~35 s, and cattle drank on average ~12 times per day, indicating that feedlot steers will consume their daily water intake in ~7 min. Thus, if there are 50 steers within a pen, the water tank can be expected to be occupied for over 5.5 h. These results emphasize the sensitivity of drinking behavior to individual history and the inflexibility of this behavior to a social mixing stressor. 

In this study, the stabilization of drinking behavior, rumination, activity durations, and eating behavior after the first week suggests that the process of familiarization with a novel environment occurs within one week for feedlot steers. Steers drank more often and had the shortest mean drinking bout duration the day after being sorted and placed into new pens irrespective of social mixing treatment, suggesting a heightened state of vigilance during times of environmental acclimation and social instability. Vigilance, or scanning behavior, allows cattle to detect a variety of stimuli [54]. This behavior evolved to allow cows to detect and avoid predators [55,56,57], and it is known to be impacted by a variety of social and environmental factors [58]. Yet, little is known regarding the impact of changes in social and environmental factors on vigilance among domesticated cattle [59] and the relationship between vigilance and drinking behavior. In adult beef cows, lactation status, group size, and the environmental landscape influence vigilance during grazing [59]. The baseline vigilance rates of adult beef cows (~3%) were observed to be lower than those of wild ungulates (i.e., bison (9.6–18.9%), African buffalo (4.2 to 5.6%), female elk (12–30%), and giraffes (18–25%)) [59], illustrating that animals that have been regularly exposed to humans are less vigilant. Habituating calves to a human presence prior to weaning may reduce a calf’s motivation to be vigilant at the feed yard.

Social mixing creates a paradigm where social facilitation can occur. Social facilitation is a phenomenon that promotes group synchrony through individuals adapting the posture or behavior of another individual within a group [60]. This was observed in the changes in eating behavior observed for the MIX cattle between week 1 and week 2. On week 1, the proportion of steers eating in the MIX pen was halfway between the proportion of steers eating from SOURCE_B and SOURCE_A. By week 2, the feeding behavior of the cattle in the MIX pens mirrored that of the cattle in SOURCE_A, suggesting that the cattle from SOURCE_A, through social facilitation, influenced the feeding behavior of the steers from SOURCE_B. These results suggest that experienced “training” animals could be effective in promoting feeding behavior. 

Herd synchrony (Shannon’s D) was lower among the SOURCE_B steers compared to that of the MIX or SOURCE_A steers. Herd synchrony provides advantages like improved protection from predators [61] and is a measure of group cohesion [62]. Synchrony is both an indicator of sufficient resource availability [62] and sociability [63,64], i.e., the motivation of an individual to remain close to conspecifics. The differences observed in this study suggest that cattle from different sources may be inherently different in their sociability. These results are supported by the reduced allogrooming, feedbunk displacements, and long headbutts observed in SOURCE_B steers. Allogrooming is a critical component of social relationship formation and maintenance among cattle, and feedbunk displacements and long headbutts are integral to social hierarchy establishment and the performance of play behaviors [65].

Social familiarization includes the learning and recognition of others as part of relationship establishment. Allogrooming is a reliable indicator of affiliative bonds between cattle [66,67], and the amount of allogrooming exchanged between two individuals is reflective of the strength of their social bond. The increases in head butting and feedbunk displacements, combined with the reduction in allogrooming, on d8 suggest that after one week, the steers had become familiar with their environment and began establishing their social structure. Social behaviors subsided by d16, suggesting that social hierarchies had been established. The divergence in rumination behavior between socially mixed SOURCE_A and socially mixed SOURCE_B steers starting on d8 suggests that steers from different sources differed in their responses to social mixing. Steers from SOURCE_B that were socially mixed experienced a drop in rumination during this period, suggesting that these individuals were either consuming less feed, spending more time being vigilant or engaging in social interactions, or were unable to sufficiently relax to allow rumination to occur. This variation in the performance of agonistic and affiliative social interactions following social mixing and/or during the first week may be an indicator of the process of familiarization with each other and the new environment, but formal social hierarchy formation may not occur until the second week. Regardless, source, not social mixing treatment, appeared to have the greatest impact on behaviors.

Though physical social interactions (i.e., agonistic and affiliative behaviors) are important in relationship establishment [67], other behaviors commonly measured as part of herd synchrony (i.e., lying, standing, and locomotion) have been found to take between three and 15 days to stabilize [9,68,69,70,71]. During the establishment of social bonds, the instability of core behaviors may be a result of cattle either adapting their daily behavioral patterns to match those of their new pen mates as part of social facilitation or spending more time engaged in various social behaviors as they establish a social hierarchy. 

Prolonged headbutts could be a sign of aggression. Juvenile cattle will engage in ‘play’ fights that can include headbutts, mounting, and head shaking. Play flights usually start with no solicitation from one of the partners and end abruptly without consequences for either individual [65]. If the observed prolonged headbutts were play behavior, then the increased engagement in both allogrooming and prolonged headbutting may indicate the higher sociability of the steers from SOURCE_A compared to those from SOURCE_B. Regardless, these cattle from different sources differed in their willingness to engage in social interactions, which may have contributed to their differing responses to social mixing.

Many of the studies that have evaluated social behavior used adult dairy cows [40]. Thus, the differences between the agonistic behavior exhibited by cattle after social mixing in this study and that observed in previous studies may be representative of the impact of cattle characteristics, like age or sex, on social behavior [72]. The steers in the present study were ~11 months old on d 0 and had not reached full maturity. Stable social hierarchal relationships are initially established when cattle are as young as four months of age [73], so these calves would have achieved a social status similar to their dams. During this stage of development, young cattle will frequently test each other’s strength [74], a process that destabilizes the social hierarchy and is characterized by an increase in the exhibition of agonistic behavior [75].

## 5. Conclusions

Socially mixing two sources of genetically similar feedlot steers after a 39 d adaptation period did not impact group-level productivity; however, the impact of social mixing was observed to vary by source. In the present study, social mixing did not impact group productivity or the exhibition of agonistic behaviors and appeared to positively facilitate feeding behavior. Yet, steer productivity varied by the treatment–source combination. Both ADG and drinking frequency were reduced in socially mixed steers of one source, whereas the other source’s steers had similar responses regardless of the mixing treatment. Thus, cattle origin and previous experiences may impact each individual animal’s behavior, productivity, and ability to cope with social stressors in a feedlot setting. This has welfare implications for cow–calf producers. The combination of early life experiences and the stage of development means that cow–calf producers can have the greatest long-term impact on cattle welfare. Ensuring that cattle have been exposed to multiple stimuli and positive human–animal interactions prior to weaning may yield long-term benefits. Cattle will become habituated to human interactions and will have experiences that prepare them to cope with contemporary stressors as they transition through the beef system.

The cattle in this study were reared and cared for within a university system. Their management, by design, was likely more similar between these sources than would be expected between sources in a commercial setting. Because management strategies are highly variable in this industry, this suggests that any differences observed in this study in response to social mixing may be larger among commercially raised cattle. 

While a direct impact between pre-weaning handling and feedlot productivity was not explicitly explored in this study, our data are indicative of potentially improved feed yard productivity if cattle are handled more frequently and for longer durations before weaning. While the different sectors of the beef industry are financially independent, the experiences that the animals have as they transition across sectors are not. This makes the case that feedlot producers have a financial interest in the management of a calf before weaning. Nonetheless, the results from this experiment indicate that social mixing has the potential to impact individual feedlot steer behavior, productivity, and welfare and emphasize the long-term impact of an animal’s source on feedlot cattle welfare and productivity.

## Figures and Tables

**Figure 1 animals-13-02981-f001:**
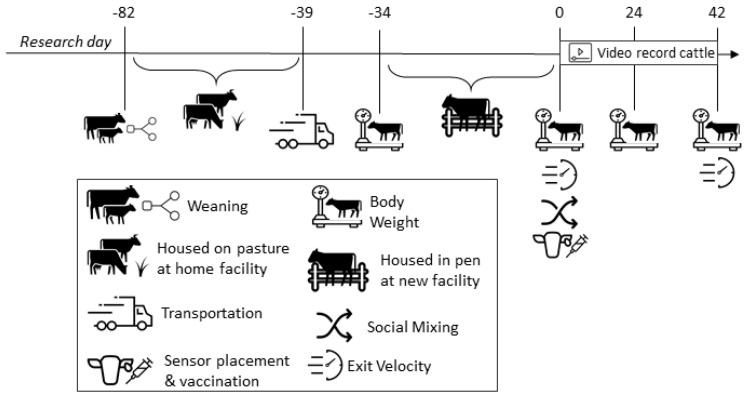
Timeline of cattle management, treatment implementation, and data collection.

**Figure 2 animals-13-02981-f002:**
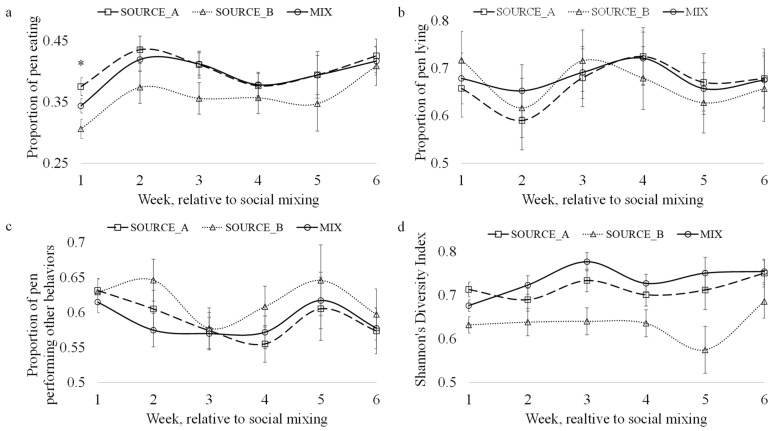
Impact of social mixing and time on the proportion of cattle in a pen observed (**a**) eating, (**b**) lying, and (**c**) performing other unspecified behaviors as well as (**d**) the Shannon–Weiner Diversity Index. Data were collected from video recordings using instantaneous scan sampling. Differences between treatments within a week are indicated with an asterisk (*).

**Figure 3 animals-13-02981-f003:**
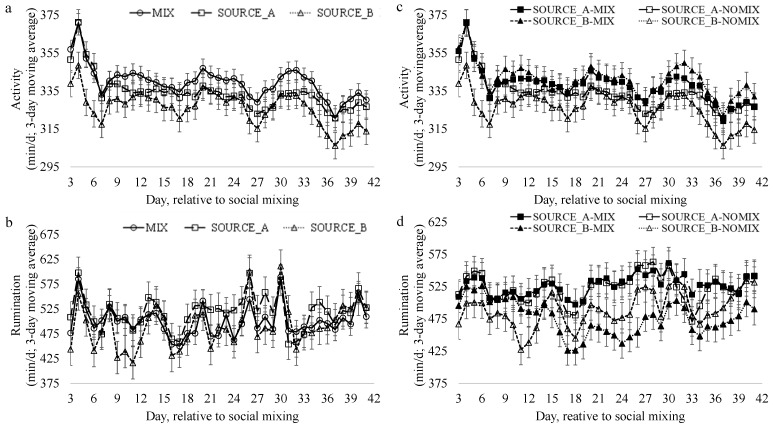
Impact of social mixing treatments on the (**a**) activity and (**b**) rumination behavior of drylot-housed steers as well as the impact of the source and social mixing treatment combination on (**c**) activity and (**d**) rumination behavior as measured via accelerometers attached to the ears of the animals. For panels (**a**,**b**), steers were housed in pens that were either socially mixed (MIX; n = 6 pens; 8 steers/pen; 50% from each source) or not socially mixed (SOURCE_A; n = 3 pens; 8 steers/pen and SOURCE_B; n = 3 pens; 8 steers/pen). For panels (**c**,**d**), steers were categorized based upon their sources and social mixing treatment experience (SOURCE_A-NOMIX; n = 24 steers, SOURCE_B-NOMIX; n = 24 steers, SOURCE_A-MIX; n = 24 steers, SOURCE_B-MIX; n = 24 steers). Data are presented as 3-day moving averages.

**Figure 4 animals-13-02981-f004:**
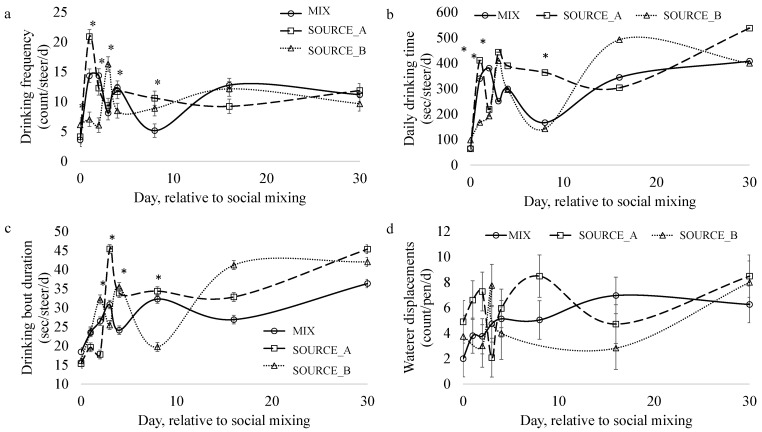
The impact of social mixing was evaluated for (**a**) drinking frequency (bout/d), (**b**) the total amount of time (sec/d) per day spent drinking for a single steer, (**c**) drinking bout duration (sec/bout), and (**d**) daily waterer displacement frequency (count/d) in pens of steers that were either socially mixed (MIX; n = 6 pens; 8 steers/pen; 50% from each source) or not socially mixed (NOMIXSOURCE_A; n = 3 pens; 8 steers/pen and NOMIXSOURCE_B; n = 3 pens; 8 steers/pen). Differences between treatments within a week are indicated with an asterisk (*).

**Figure 5 animals-13-02981-f005:**
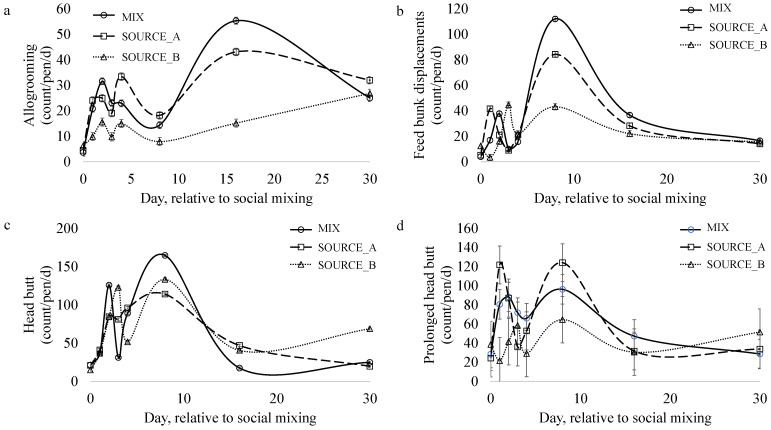
The impact of social mixing on the frequency of (**a**) allogrooming bouts, (**b**) feedbunk displacements, (**c**) head butts, and (**d**) long headbutts performed per day in pens of steers that were either socially mixed (MIX; n = 6 pens; 8 steers/pen; 50% from each source) or not socially mixed (SOURCE_A; n = 3 pens; 8 steers/pen and SOURCE_B; n = 3 pens; 8 steers/pen). Data were collected from video recordings using continuous observations.

**Table 1 animals-13-02981-t001:** Composition and nutritional profile of the standard grower ration fed to the cattle throughout the study.

Item	% of Diet (Dry Matter Basis)
Ingredient	
Sorghum–Sudan grass hay	17.00%
Corn stalks	10.00%
Steamed flaked corn	36.15%
Sweet Bran	28.25%
Pellet Supplement	3.50%
Urea	0.75%
Corn oil	3.80%
Limestone	0.55%
Nutrient content	
NE_m_, Mcal/kg NE_g_, Mcal/kg	0.81 0.52
Total Digestible Nutrients, %	73.7
Rumen Degradeable Protien, %	17.4
Fat, %	6.3
Ca:P ratio	1.31

Steers were blocked according to their sources and stratified by d −34 body weight before being randomly assigned to one of twelve pens (n = 8 steers/pen). Each pen was assigned to one of three treatments: (1) SOURCE_A—100% of cattle from SOURCE_A (n = 3 pens; n = 8 steers/pen), (2) SOURCE_B—100% of cattle from SOURCE_B (n = 3 pens; n = 8 steers/pen), and (3) MIX—50% of cattle from SOURCE_A and 50% of cattle from SOURCE_B (n = 6 pens; n = 8 steers/pen).

**Table 2 animals-13-02981-t002:** Ethogram of feedlot cattle behavior recorded during behavior observations.

Behavior	Definition
Eating	Steer has its head in a feed bunk.
Drinking	Steer has its head in a water trough and appears to be swallowing.
Lying	Steer is recumbent, i.e., not supported by legs (lying laterally or sternally).
Standing	Steer is standing on four legs without locomotion.
Allogrooming	Licking movements carried out by one steer on the body of another, characterized by repetitive back-and-forth head movements performed by the actor in direct contact with the reactor. Once the actor stops grooming the reactor for more than 10s, the bout is finished.
Headbutting	Head of one steer connects with the body of another steer.
Prolonged headbutt	A prolonged headbutt was recorded when the actor initiated head-to-head or head-to-neck contact with another steer that lasted longer than 5s. It is characterized by continuous direct contact between both steers, wherein steers potentially pushed or maneuvered against each other. When steers lost contact for more than 10s, the bout was finished.
Mounting	Steer positions body on the top of another steer’s topline
Feed bunk displacement	A displacement was recorded when a butt or a push from the “actor” resulted in the complete withdrawal of the head of another individual (the “recipient”) from the feed bunk ^1^_._
Waterer displacement	A displacement was recorded when a butt or a push from the “actor” resulted in the complete withdrawal of the head of another individual (the “recipient”) from the waterer ^1^_._

^1^ Adapted from [6].

**Table 3 animals-13-02981-t003:** Group mean and standard errors (±SEM) of productivity responses of feedlot cattle that were socially mixed (MIX; n = 6 pens; 8 steers/pen; 50% of steers from SOURCE_B and 50% of steers from SOURCE_A) or not socially mixed (NOMIX-SOURCE_A; n = 3 pens; 8 steers/pen and NOMIX-SOURCE_B; n = 3 pens; 8 steers/pen).

Item	MIX	SOURCE_B	SOURCE_A	SEM	*p*-Value
ADG ^1^, kg	1.50	1.57	1.59	0.05	0.59
DMI ^2^, kg	10.72	10.58	10.78	0.32	0.48
G:F ^3^	0.14	0.15	0.14	0.04	0.48

^1^ Average daily gain (ADG) was calculated using body weights (BW) collected on d 0, 24, and 42; ADG for each period was calculated as the difference in BW between the two respective time points divided by the number days within that time period. ^2^ Steer dry matter intake (DMI) for each period was calculated by dividing the total daily amount of feed delivered (kg of dry-matter) to each pen by the number of steers within each pen and the number of days within that time period. ^3^ Steer G:F for each period was calculated by dividing the total BW gain (kg) for all steers within a pen and the total DMI for a pen by the number of steers within each pen.

**Table 4 animals-13-02981-t004:** Impact of the source–social mixing treatment combination on mean (±SEM) steer behavior (e.g., rumination, activity, and drinking) and productivity. Steers were categorized based upon their source herds and social mixing treatment experiences (SOURCE_A-NOMIX; n = 24 steers, SOURCE_B-NOMIX; n = 24 steers, SOURCE_A-MIX; n = 24 steers, SOURCE_B-MIX; n = 24 steers).

Item	SOURCE_B MIX	SOURCE_BNOMIX	SOURCE_AMIX	SOURCE_ANOMIX	SEM	*p*-Value
Drinking Behavior ^1^						
Bout frequency, count/steer	8.04 ^a^	8.30 ^a^	10.52 ^b^	9.93 ^b^	1.08	0.04
Total time, sec/steer	223.65	244.06	276.88	299.26	1.10	0.80
Mean bout duration, sec/bout/steer	27.89	29.38	26.50	30.13	1.07	0.48
ADG ^2^, kg	1.41	1.59	1.60	1.57	0.039	0.10
EV ^3^, m/s	2.45 ^a^	2.34 ^a^	1.26 ^b^	1.49 ^b^	0.16	<0.0001
Rumination ^4^, min/steer/day	477	487	527	523	19.44	0.21
Activity ^4^, min/steer/day	340.89	326.09	338.15	334.00	9.12	0.41

^a,b^ Unique superscripts differ within a row when *p* < 0.05. ^1^ Continuous observations made from 0800 to 1800 h measured the number of social behaviors and drinking bouts exhibited and engaged in by each individual within a pen on d 1, 2, 3, 4, 8, and 16 relative to social mixing. ^2^ Average daily gain (ADG) was calculated using body weights (BW) collected on d 0, 24, and 42; ADG for each period was calculated as the difference in BW between the two respective time points divided by the number days within that time period. ^3^ Steer EV, as described by [19], was measured on d 0; EV was calculated as follows: 1.8 m divided by the time taken to travel 1.8 m. ^4^ Rumination and activity durations were recorded continuously from d 0 to d 42 using an ear tag containing a three-axis accelerometer.

**Table 5 animals-13-02981-t005:** Group mean (± SEM) behavioral responses of feedlot cattle over time that were either socially mixed (MIX; n = 6 pens; 8 steers/pen; 50% from each source) or not socially mixed (SOURCE_A; n = 3 pens; 8 steers/pen and SOURCE_B; n = 3 pens; 8 steers/pen).

	Treatment				
Item	MIX	SOURCE_B	SOURCE_A	SEM	*p*-Value
EV ^1^, m/s	1.86 ^a^	2.34 ^b^	1.48 ^a^	0.22	0.02
Herd behavior ^2^, proportion of pen					
Lying	0.68	0.67	0.67	0.02	0.614
Eating	0.39 ^a^	0.36 ^b^	0.40 ^a^	0.01	0.005
Other	0.59	0.62	0.59	0.01	0.059
Shannon–Weiner Diversity Index ^3^	0.72 ^a^	0.64 ^b^	0.72 ^a^	0.01	<0.0001
Social Behavior ^4^					
Allogrooming (bout/pen/d)	19.97 ^a^	12.09 ^b^	21.34 ^a^	0.13	0.000
Feedbunk displacement (count/pen/d)	20.13	17.76	20.29	0.25	0.936
Waterer displacement (count/pen/d)	4.43	4.73	5.48	0.17	0.581
Headbutt (count/pen/d)	45.80	52.67	56.86	0.17	0.709
Long headbutt (count/pen/d)	63.37	41.94	63.88	7.50	0.090
Mounting (count/pen/d)	4.92	4.32	4.16	0.21	0.764
Rumination duration ^5^, min/d	500.24 ^b^	489.97 ^b^	518.98 ^a^	31.6	0.0004
Activity duration ^5^, min/d	342.03 ^a^	327.74 ^b^	336.00 ^a^	4.43	<0.0001
Drinking behavior ^4^					
Daily bout frequency, count/steer	9.26	8.84	10.36	0.11	0.782
Total daily time drinking, count/steer	249.44	243.81	298.75	0.09	0.302
Mean daily bout duration, sec/bout	26.84	28.03	28.39	0.09	0.639

^a,b^ Unique superscripts differ within a row when *p* < 0.05. ^1^ Steer EV, as described by Burrow and Dillion (1997), was measured on d 0; EV was calculated as follows: 1.8 m divided by the time taken to travel 1.8 m. ^2^ Instantaneous scan sampling conducted from 0800 to 1800 h at 10 min intervals and was used to assess the proportions of steers in each pen that were lying, eating, or performing other behaviors on d 1, 2, 3, 4, 5, 6, 7, 8, 9, 14, 15, 16, 21, 22, 23, 28, 29, 30, 35, 36, and 37 relative to social mixing. ^3^ The Shannon–Weiner Diversity Index [17] was calculated for each 10 min instantaneous scan sample observation for each pen and then averaged by treatment. ^4^ Continuous observations made from 0800 to 1800 h were used to assess the number of social behaviors and drinking bouts exhibited and engaged in by each individual within a pen on d 1, 2, 3, 4, 8, and 16 relative to social mixing. ^5^ Rumination and activity were recorded continuously from d 0 to d 42 using an ear tag containing a three-axis accelerometer.

## Data Availability

The data presented in this study are available on request from the corresponding author.

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
