# Peer review of "Consider the Source: The Impact of Social Mixing on Drylot Housed Steer Behavior and Productivity"

_animals, 2023, doi:10.3390/ani13182981_

Round 1

Reviewer 1 Report

An interesting read - there were considerations that I had not previously made relative to cattle but have similar considerations with other species.  While reading, I did have questions arise and not certain that they were adequately answered.  

I will note that I felt that the Introduction can be cleaner – it was a bit redundant at times and the order, or presentation of concepts, could be streamlined to keep "like concepts with like concepts".  To me, it read a little "back and forth".

Line 410 reads “The results of this research emphasize the importance of assessing welfare at the individual level – as the impact of a social mixing treatment on cattle welfare differed depending on the metric evaluated and their source herd of origin.”   If cattle are individuals (which, of course they are) and individuals differ, then predication on adverse impact is not possible – since “some but not all” cattle will be negatively impacted by social mixing and knowing “which” individuals would be negatively impacted can’t be known…… (correct????)      Follow Up:  Lines 422- 426 helps to clarify for me.....

Note to myself:  Lines 581 – 582  "Thus, cattle origin and previous experiences may impact each individual animal’s behavior, productivity, and their ability to cope with social stressors in a feedlot setting” = for me, this is “key” and I would prefer to have this finding emphasized and discussed more.   That is, my personal preference is see to more discussion on the connection between the findings and the "animal welfare" part; i.e. "why does this finding matter?"  "what implications for improving animal welfare follow from the finding?"  "can/should the finding(s) be included in policy, procedures, or for other change?"

Lastly, in the Abstract, it is stated, "Social mixing was not universally detrimental to cattle welfare, and source considerations should be made when implementing this management practice".    I do seem to be missing the answer to the "Why?" question - WHY should considerations be made?  What does this do for animal welfare?  

Admittedly, I found some of the compound sentence structure cumbersome in that I had to re-read for clarity and understanding.

Quick suggestion:  Line 56 reads:  “access to and use of resources (e.g., feeder, water) = correct this to “access to, and use of, resources….”

Author Response

An interesting read - there were considerations that I had not previously made relative to cattle but have similar considerations with other species.  While reading, I did have questions arise and not certain that they were adequately answered.  

I will note that I felt that the Introduction can be cleaner – it was a bit redundant at times and the order, or presentation of concepts, could be streamlined to keep "like concepts with like concepts".  To me, it read a little "back and forth".

 AU: thank you for this observation.  We have streamlined and revised the introduction.

Line 410 reads “The results of this research emphasize the importance of assessing welfare at the individual level – as the impact of a social mixing treatment on cattle welfare differed depending on the metric evaluated and their source herd of origin.”   If cattle are individuals (which, of course they are) and individuals differ, then predication on adverse impact is not possible – since “some but not all” cattle will be negatively impacted by social mixing and knowing “which” individuals would be negatively impacted can’t be known…… (correct????)      Follow Up:  Lines 422- 426 helps to clarify for me.....

AU: Thank you for this observation.  We have elaborated upon this in lines 442-452

Note to myself:  Lines 581 – 582  "Thus, cattle origin and previous experiences may impact each individual animal’s behavior, productivity, and their ability to cope with social stressors in a feedlot setting” = for me, this is “key” and I would prefer to have this finding emphasized and discussed more.   That is, my personal preference is see to more discussion on the connection between the findings and the "animal welfare" part; i.e. "why does this finding matter?"  "what implications for improving animal welfare follow from the finding?"  "can/should the finding(s) be included in policy, procedures, or for other change?"

AU: we have expanded the conclusion and revised the discussion to address this point.  Thank you for the opportunity to expand this explanation.

Lastly, in the Abstract, it is stated, "Social mixing was not universally detrimental to cattle welfare, and source considerations should be made when implementing this management practice".    I do seem to be missing the answer to the "Why?" question - WHY should considerations be made?  What does this do for animal welfare?  

AU: Thank you for this observation.  We have revised the abstract as such: “Group analyses indicate that socially mixing two sources of feedlot steers did not negatively impact group productivity; yet the impacts that were observed at the individual level suggest that prior experiences may influence their ability to cope with social stress emphasizing the importance of early life experiences on long-term welfare and productivity. Social mixing was not universally detrimental to cattle welfare, and source may have the greatest affect on their performance regardless of a social mixing event.”

Reviewer 2 Report

General comments: This manuscript includes performance and behavioral data from 96 crossbred steers of similar genetic composition sourced from two rearing locations and transported to a feedlot after weaning and backgrounding.  After an adaptation period at the feedlot, steers were placed in pens with others from the same source (3 pens each) or in pens mixed with steers from both sources (6 pens total) and equipped with monitoring technology. For much of the analyses, pen is considered the experimental unit and data are analyzed accordingly. Data on individual steers are also summarized to add perspective to the study but may be less reliable. The data are of interest and the experiment is designed to elucidate effects of mixing cattle without other confounding factors typical of commercially comingled cattle. Overall, the manuscript is generally well written with some exceptions as noted below.

Figure 1 is well done in showing the timeline for the study. Figures 2, 3, 4, and 5 are complex and may be subject to differing interpretations but do add to the value of the manuscript.

Specific comments:

L16: “… did not affect pen average …”

L17-18: “… negatively affected productivity of steers from one source but not from the other source.”

L30: Use “whereas” rather than “while.” Also lines 49, 440, 581, 592.

L68: “Cattle from sale barn auctions …”

L94-95: Not sure why SE are used unless representing multiple days? THI ranged from XX to XX.

L 100 and several other places: Use “before” or “Before” rather than “prior to”

L118: Use actual Km distances from each location – no SE needed.

L120: “Steers were housed separately by source without …”

L 161: “All behavior data were collected …” Also lines 285, 290, 362, 390.

L166: ”among observers”

Formatting issues with Tables 1 and 2.

L170: Use plural: “Statistical analyses”

L275-276: Not clear; reword for clarity.

L321: “asterisk” rather than “asketerisk”

More of an explanation/justification is needed for why 3-day moving averages were used in Figure 3.

L467: Use “though” rather than “while” Also lines 546 and 589.

L590: “ … this study, our data are indicative of potentially improved feedyard productivity if cattle are handled more frequently and for longer duration before weaning.”

Overuse of "prior to" "while" and perhaps "impact" compared to other wording options.

Author Response

General comments: This manuscript includes performance and behavioral data from 96 crossbred steers of similar genetic composition sourced from two rearing locations and transported to a feedlot after weaning and backgrounding.  After an adaptation period at the feedlot, steers were placed in pens with others from the same source (3 pens each) or in pens mixed with steers from both sources (6 pens total) and equipped with monitoring technology. For much of the analyses, pen is considered the experimental unit and data are analyzed accordingly. Data on individual steers are also summarized to add perspective to the study but may be less reliable. The data are of interest and the experiment is designed to elucidate effects of mixing cattle without other confounding factors typical of commercially comingled cattle. Overall, the manuscript is generally well written with some exceptions as noted below.

Figure 1 is well done in showing the timeline for the study. Figures 2, 3, 4, and 5 are complex and may be subject to differing interpretations but do add to the value of the manuscript.

 AU: Thank you.

Specific comments:

L16: “… did not affect pen average …”

AU: corrected

L17-18: “… negatively affected productivity of steers from one source but not from the other source.”

AU: corrected

L30: Use “whereas” rather than “while.” Also lines 49, 440, 581, 592.

AU: corrected

L68: “Cattle from sale barn auctions …”

AU: corrected

L94-95: Not sure why SE are used unless representing multiple days? THI ranged from XX to XX.

AU: revised for clarity.

L 100 and several other places: Use “before” or “Before” rather than “prior to”

AU: corrected

L118: Use actual Km distances from each location – no SE needed.

AU: corrected

L120: “Steers were housed separately by source without …”

AU: corrected

L 161: “All behavior data were collected …” Also lines 285, 290, 362, 390.

AU: corrected

L166: ”among observers”

AU: corrected

Formatting issues with Tables 1 and 2.

AU: corrected

L170: Use plural: “Statistical analyses”

AU: corrected

L275-276: Not clear; reword for clarity.

AU: revised for clarity

L321: “asterisk” rather than “asketerisk”

AU: corrected

More of an explanation/justification is needed for why 3-day moving averages were used in Figure 3.

AU: corrected and text added to the M&M to address this point.

L467: Use “though” rather than “while” Also lines 546 and 589.

AU: corrected

L590: “ … this study, our data are indicative of potentially improved feedyard productivity if cattle are handled more frequently and for longer duration before weaning.”

AU: corrected